# Towards Improving Calibration in Object Detection Under Domain Shift

**Muhammad Akhtar Munir**[1,2*]**, Muhammad Haris Khan**[2]**, M. Saquib Sarfraz**[3,4]**, Mohsen Ali**[1]

[1] Information Technology University of Punjab, [2] Mohamed bin Zayed University of Artificial Intelligence,
[3] Karlsruhe Institute of Technology, [4]Mercedes-Benz Tech Innovation

## Abstract

With deep neural network based solution more readily being incorporated in real-world applications, it has been pressing requirement that predictions by such models, especially in safety-critical environments, be highly accurate and well-calibrated. Although some techniques addressing DNN calibration have been proposed, they are only limited to visual classification applications and in-domain predictions. Unfortunately, very little to no attention is paid towards addressing calibration of DNN-based visual object detectors, that occupy similar space and importance in many decision making systems as their visual classification counterparts. In this work, we study the calibration of DNN-based object detection models, particularly under domain shift. To this end, we first propose a new, plug-and-play, train-time calibration loss for object detection (coined as TCD). It can be used with various application-specific loss functions as an auxiliary loss function to improve detection calibration. Second, we devise a new implicit technique for improving calibration in self-training based domain adaptive detectors, featuring a new uncertainty quantification mechanism for object detection. We demonstrate TCD is capable of enhancing calibration with notable margins (1) across different DNN-based object detection paradigms both in in-domain and out-of-domain predictions, and (2) in different domain-adaptive detectors across challenging adaptation scenarios. Finally, we empirically show that our implicit calibration technique can be used in tandem with TCD during adaptation to further boost calibration in diverse domain shift scenarios.

## 1 Introduction

Owing to the success of deep neural network in last decade, they have been progressively becoming part of different safety-critical applications, including autonomous driving [5], healthcare [2, 32], and legal research [40]. In such high-stake applications, it is of paramount importance that the model predictions are not only correct, but also well-calibrated. For a calibrated model, probability of being correct across all confidence levels and the predictions-confidence should be aligned [6, 35] It is desired that the incorrect predictions have low-confidence, however, a poorly calibrated model is prone to have confident but incorrect predictions.

We have witnessed tremendous effort towards improving the predictive accuracy of models, however, considerably less attention is devoted to model calibration. Among a few works addressing model calibration, the majority of them focus on classification tasks [6, 17, 20, 22, 35]. A dominant class of methods address calibration by proposing different post-hoc approaches [28, 6], The core idea is to transform model outputs by a single parameter which is optimized on a validation set, with

---

*Corresponding author, Intelligent Machines Lab, Department of Computer Science, Information Technology University of the Punjab, Lahore, Pakistan. Email: `akhtar.munir@itu.edu.pk` Project Page: `http://im.itu.edu.pk/towards-improving-calibration/`

36th Conference on Neural Information Processing Systems (NeurIPS 2022).

an objective to improve the calibration of in-domain predictions. Such methods involve limited parameters while calibrating the outputs of the models. Further, they are restrictive, since in many real-world scenarios, a validation set is not always available. To involve all model parameters, some methods propose train-time calibration techniques [17, 20, 22] by constraining that for a given sample, the predicted class confidence and likelihood for that class should be minimized. The model could remain poorly calibrated for the classes with non-maximum prediction confidences [7].

Surprisingly, little to no attention has been paid towards addressing the calibration of deep learning based visual object detection methods, that form an important part of many decision making systems. Also, most current efforts aim at improving model calibrations only for in-domain predictions. However, in many real-world scenarios, owing to domain shift, the distribution of the data received by a deployed model can be very different to the distribution of its training data. Therefore for several practical scenarios, a model should be well-calibrated for both in-domain and out-of-domain predictions. Besides carrying scientific value, well-calibrated object detectors, especially under domain drift, will substantially improve overall trust in many vision-based safety-critical applications and will be of great value to industry practitioners.

In this paper, we study the calibration of object detection models for both in-domain and out-of-domain detections. We observe that: (1) detection models demonstrate poor calibration for in-domain and out-of-domain detections (see Fig. 1), and (2) unsupervised domain adaptive detection models are rather miscalibrated when compared to their predictive accuracy in a target domain.

Towards developing well-calibrated object detection models, for both in-domain and out-of-domain scenarios, we propose a new plug-and-play loss formulation, termed as *train-time calibration for detection* (TCD). It can be used with task-specific loss functions during training phase and acts as a regularization for detections. In addition, we develop an implicit calibration technique for self-training based domain adaptive detectors. Finally, we empirically show that this technique is complementary to our loss function and they both can be utilized during adaptation to further boost calibration under challenging domain shift detection scenarios. We validate the effectiveness of our loss function towards improving calibration of different DNN-based object detection paradigms and different domain adaptive detection models under challenging domain shift scenarios.

## 2   Related Work

Below we survey different techniques in literature aimed at addressing the calibration of deep neural networks. Broadly, the existing calibration techniques can be divided into post-hoc and train-time calibration methods. The former estimates a transformation through a hold-out data after training, whereas the latter involves model parameters during the training. Further, there are methods that achieve implicit calibration through using model's uncertainty or learning to reject OOD samples.

**Post-hoc calibration methods:** Post-hoc calibration methods are post processing methods that leverage a hold-out validation data to optimize the calibration parameters for transforming the outputs of trained deep neural networks. A popular post-hoc calibration approach is temperature scaling (TS) [6] (an extension of Platt Scaling [28]) that re-scales the logits by an optimal temperature ($T$) parameter which is estimated on a validation set. The overall accuracy of the model remains unaffected by temperature scaling, but the logits values are modulated in such a way that confidence scores are reduced. Temperature scaling along with other methods like vector scaling and matrix scaling extends Platt scaling to multiclass setting. [11] introduced bin wise temperature scaling that incorporates different temperatures for different confidence intervals. To mitigate miscalibration under long-tailed distribution scenarios, Islam et al. [10] used class frequency information. Specifically, the model is calibrated using class distribution-aware temperature scaling and label smoothing. The Dirichlet calibration (DC) [15] is derived from Dirichlet distributions and extends the beta calibration method [16] to multi-class setting. This is achieved by training an extra neural network layer, that inputs log-transformed class probabilities, over the hold-out validation data. Recently, [35] generalizes existing post-hoc calibration methods, to address the poor calibration of out-of-domain predictions, by applying post-hoc calibration step after transforming validation set. In general, post-hoc methods for model calibration require hold-out validation set. Also, the majority of post-hoc methods perfrom model calibration for in-domain predictions.

**Train-time Calibration methods:** Negative log-likelihood (NLL) is the most popular approach to train a DNN based classifier. Recent work shows that NLL-based training results in overconfident

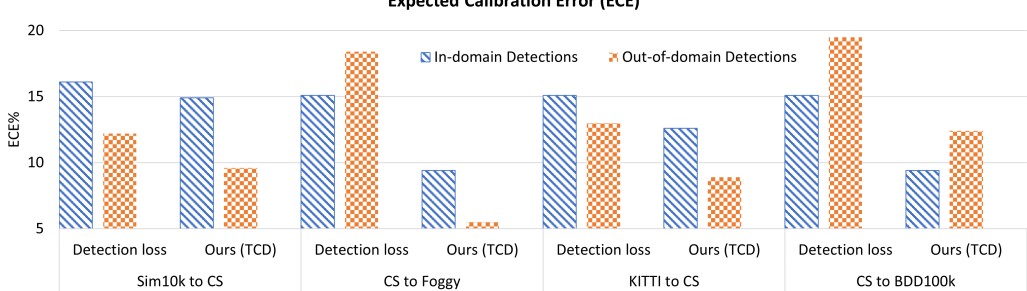

(a) Calibration performance (ECE) of a DNN-based detector (FCOS [34]) trained using task-specific detection loss and our method (task-specific detection loss+TCD), where TCD is the proposed auxiliary loss.

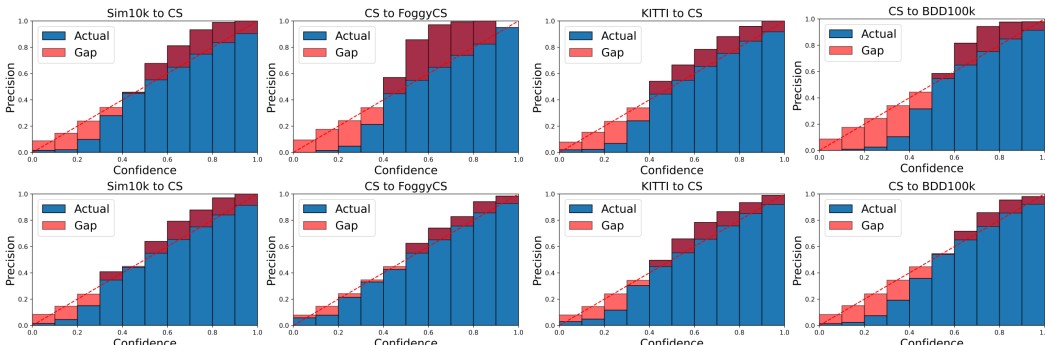

(b) Reliability diagrams. Top row: DNN-based detector (FCOS [34]) trained using task-specific loss. Bottom row: Ours, trained with adding the proposed TCD loss.

Figure 1: DNN-based object detectors trained with our proposed calibration loss results in better calibration.

scores [6]. To this end, various methods presented additional loss formulations that can be used with task-specific losses during the training procedure. For instance, [27] penalized overconfident predictions by adding the negative of entropy to the loss function. Similarly, Liang et al. [20] improved calibration by proposing the difference of accuracy and confidence as an auxiliary loss to the cross-entropy loss. Whereas, [17] proposed an auxiliary loss, for obtaining improved calibration, which is quantified with the help of reproducing kernel in a Hilbert space. Recently, [7] developed an auxiliary loss formulation for calibrating non-predicted labels along with the predicted one. It has been shown that methods which attempt to reduce overconfidence by modifying the hard-labels can achieve better model calibration. Mukhoti et. al. [23], hypothesized that minimizing focal loss [21] increases the predicted distribution's entropy while trying to minimize KL divergence between the predicted and ground-truth distribution. Replacing cross entropy with Focal loss results in a well-calibrated classification model. [24] empirically showed that label smoothing (LS), originally proposed by [33], helps in improving calibration. [14] calibrated uncertainty by relating accuracy and uncertainty, that yields accurate predictions to be more certain while inaccurate predictions to be more uncertain.

**Other relevant methods:** A recent work [37] based on empirical results proposed a unified framework that includes training and post-hoc calibration for better calibration. Hein et al. [8] identified that ReLU activation (or similar piece-wise linear functions) is a core reason behind the overconfident predictions of DNNs far away from the training data. To overcome this, the data augmentation is incorporated using adversarial learning. [13] detected OOD samples by performing spectral analysis over initial CNN layers, resulting in much better calibrated model. Finally, some works report implicitly calibrating DNNs via some loss function [24] or selecting pseudo-labels during self-training based on model's uncertainty [25].

Note that, almost all aforementioned techniques for analyzing and improving DNN calibration focus on classification tasks. There is little to no literature studying the calibration of DNN-based object

detectors, including under domain shift. Toward this aim, inspired by the train-time calibration methods, we propose a new auxiliary loss formulation (TCD) for DNN-based object detection calibration that can be integrated with any task-specific loss function while training. It is capable of significantly calibrating in-domain and out-of-domain object detections. In addition, we develop an implicit calibration technique, that is also complementary to our loss formulation, to boost the calibration of self-training based domain adaptive object detectors.

# 3 Improving Calibration in Object Detectors

## 3.1 Definitions of Calibration

**Calibration for classification:** Let $\mathcal{D} = \langle(\mathbf{x}_i, y_i^*)\rangle_{i=1}^N$ be a dataset of $N$ image and ground-truth pairs from a joint distribution $\mathcal{D}(\mathcal{X}, \mathcal{Y})$. Where $\mathbf{x}_i \in \mathbb{R}^{H \times W \times C}$ is an image, and $y_i^* \in \mathcal{Y} = 1, 2, ..., K$ is the associated ground-truth class label. $H$, $W$, and $C$ represent the width, height and the number of channels of an image respectively. For a classification model $\mathcal{F}_{cls}$, that predicts a label $\hat{y}$ with a confidence score $\hat{s}$, we can define a perfect calibration as [6]:

$$\mathbb{P}(\hat{y} = y^*|\hat{s} = s) = s \quad \forall s \in [0, 1]. \tag{1}$$

Where $\mathbb{P}(\hat{y} = y^*|\hat{s} = s)$ is the accuracy for a particular confidence score $s$. For the model to be calibrated, this accuracy should match with the predicted confidence.

**Calibration for object detection:** For training an object detection model, the ground-truth annotations include object's localization information along with their associated categories. Let $\mathbf{b}^* \in \mathcal{B} = [0, 1]^4$ be the bounding box annotation of the object and $y^*$ be the corresponding class label. We assume that an object detection model $\mathcal{F}_{det}$ predicts the object location $\hat{\mathbf{b}}$ and the class label $\hat{y}$ with confidence $\hat{s}$. This prediction is accurate, if the Intersection-over-Union (IoU) between $\hat{\mathbf{b}}$ and $\mathbf{b}^*$ is larger than some threshold $\gamma$, and $\hat{y}$ is same as $y^*$, i.e. $\mathbb{1}[\text{IoU}(\hat{\mathbf{b}}, \mathbf{b}^*) \geq \gamma]\mathbb{1}[\hat{y} = y^*]$. Then, for object detection, a perfect calibration can be expressed as [19] [2]:

$$\mathbb{P}(U = 1|\hat{s} = s) = s \quad \forall s \in [0, 1]. \tag{2}$$

Where $U = 1$ denotes a correctly classified prediction.

**Expected calibration error for classification and object detection:** For a classification model, the absolute difference between the confidence score and the accuracy corresponding to that confidence score (eq. 2) is considered *Calibration Error*. The miscalibration of the model is quantified by computing the expectation of calibration error over predicted confidence $\hat{s}$ [26, 6, 18].

$$\mathbb{E}_{\hat{s}}\left[|\mathbb{P}(\hat{y} = y^*|\hat{s} = s) - s|\right] \tag{3}$$

The continuous confidence space of $\hat{s}$ is divided into $M$ equally spaced bins to approximate ECE:

$$\text{ECE} = \sum_{m=1}^{M} \frac{|I(m)|}{|\mathcal{D}|} |\text{acc}(m) - \text{conf}(m)|, \tag{4}$$

where $I(m)$ is the set of samples in $m^{th}$ bin, and $|\mathcal{D}|$ is the total number of samples. $\text{acc}(m)$ and $\text{conf}(m)$ denote the average accuracy and average confidence in $m^{th}$ bin, respectively. Along similar lines, ECE for the object detection (D-ECE) can be defined as the expected deviation of precision from the predicted confidence [19]:

$$\mathbb{E}_{\hat{s}}\left[|\mathbb{P}(U = 1|\hat{s} = s) - s|\right] \tag{5}$$

As in classification ECE, confidence space is divided into $M$ equally spaced bins to approximate D-ECE :

---

[2]Note that, Eq.( 2) can also be extended to location-dependent calibration.

$$\mathrm{D-ECE} = \sum_{m=1}^{M} \frac{|I(m)|}{|\mathcal{D}|} \left| \mathrm{prec}(m) - \mathrm{conf}(m) \right|, \tag{6}$$

where $\mathrm{prec}(m)$ denotes the average precision in a bin.

## 3.2 Train-time Calibration for Detection: TCD

In this section we detail our proposed train-time calibration mechanism for object detection. It features a novel auxiliary loss function that involves both the classification and localization components. DNN-based object detectors predict a bounding box and corresponding class confidences for a detected region. In order to achieve detection calibration, it is imperative to take into account both of them. Our core idea is to jointly calibrate the estimated (class-wise) confidences and predicted bounding boxes. To achieve this, we propose to compute two quantities over a mini-batch during training: (1) the difference between the classification accuracy and the confidence, and (2) the deviation between the predicted bounding box overlap and the predicted class confidence. Specifically, there are two components in our loss formulation. Inspired by the train-time calibration techniques for classification, we adapt the confidence calibration loss [7] and develop the first component $\mathrm{d_{cls}}$. It measures the absolute difference between the average confidence and average accuracy.

$$\mathrm{d_{cls}} = \frac{1}{K} \sum_{k=1}^{K} \left| \frac{1}{L \times R} \sum_{l=1}^{L} \sum_{r=1}^{R} s_{l,r}[k] - \frac{1}{L \times R} \sum_{l=1}^{L} \sum_{r=1}^{R} q_{l,r}[k] \right|. \tag{7}$$

Where $R$ is the number of locations in the output class confidences channel map and $L$ is number of images in mini-batch. $q_{l,r}[k] = 1$ if label $k$ is the ground-truth label for sample $l$ in location $r$, and 0 otherwise. $s_{l,r}[k]$ denotes $k_{th}$ class confidence for $r_{th}$ location in sample $l$. The second component $\mathrm{d_{det}}$ computes the mean of the absolute difference between the bounding box overlap (with the ground-truth/pseudo ground-truth) and the confidence of the predicted class over all the positive regions. Let $N_{pos}^{l}$ denotes the number of positive regions in the sample $l$. The loss is defined as:

$$\mathrm{d_{det}} = \frac{1}{L} \sum_{l=1}^{L} \frac{1}{N_{pos}^{l}} \sum_{n=1}^{N_{pos}^{l}} \left| [\mathrm{IoU}(\hat{\mathbf{b}}_n, \mathbf{b}_n^*) - \hat{s}_n] \right| \tag{8}$$

$$\mathcal{L}_{\mathrm{TCD}} = \frac{1}{2}(\mathrm{d_{cls}} + \mathrm{d_{det}}) \tag{9}$$

The $\mathrm{d_{cls}}$ component in the proposed loss is capable of not only calibrating the confidence of the predicted class but also for the non-predicted classes. It penalizes the model, if for a given class $k$, the average confidence across mini-batch samples and possible output locations deviates from the average occurrence across mini-batch of this class. On the other hand, the $\mathrm{d_{det}}$ component penalizes the deviation between the IoU score (computed between the ground-truth bounding box and predicted one) and its corresponding predicted confidence for positive regions. It explicitly forces the object detector to match the confidence, of the predicted class, with the tightness of the predicted bounding-box over the detected object. Since above components are not based on binning, like Eqs.(4,6), as such, it avoids the non-differentibility issue

Note that, both loss components, $\mathrm{d_{det}}$ and $\mathrm{d_{cls}}$, operate over the mini-batch constructed during training and so the $\mathcal{L}_{\mathrm{TCD}}$ can be used as an auxiliary train-time calibration loss for calibrating object detectors, including domain-adaptive ones, in conjunction with various task-specific loss functions (secs. 4.1, 4.2). Further, we show that the proposed loss formulation can also be used with the pseudo-labels used in self-training based unsupervised domain-adaptive detection algorithms for improving calibration in target domain (sec.4.2). Finally, we show that our loss formulation is complementary to implicit calibration techniques and so can be deployed to further enhance calibration (sec.4.3).

## 3.3 Implicitly Calibrating Self-Training based Domain Adaptive Detectors

We propose a technique aimed at implicitly improving the calibration of self-training based domain adaptive detectors. Mostly, self-training based domain adaptive detectors involve constructing pseudo instance-level labels (termed pseudo-labels thereafter) corresponding to detections in a target domain. Based on these pseudo-labels, pseudo-targets are formed for computing (classification) loss during the adaptation phase. We observe that, these pseudo-targets are constructed as one-hot encoded channels, and so they struggle to reflect the predictive confidence or uncertainty of detections. As a result, the adaptation model fails to account for the noise in detections, mostly prevalent under domain shift, and could inadvertently learn to make overconfident predictions, hence negatively affecting calibration. To this end, we first present a new uncertainty quantification mechanism for object detection and then leverage this to modulate the one-hot encoded pseudo-targets.

**Quantifying uncertainty in object detection:** To quantify model's uncertainty for detections, we follow a three-step process [25]. First, given an arbitrary image, we perform $N$ stochastic forward passes (inferences) over the one-stage object detector using Monte-Carlo dropout [3]. Specifically, we apply spatial MC-dropout [36] over the convolutional filters after the feature extraction layer [25]. Let $\hat{\mathbf{b}}_{n,m} \in \mathbb{R}^4$ be the $m_{th}$ predicted bounding box during $n_{th}$ inference, $\mathbf{s}_{n,m}$ be the corresponding confidence vector, and $\hat{c}_{n,m}$ be the predicted class corresponding to the highest confidence $\hat{s}_{n,m} \in \mathbf{s}_{n,m}$. We define $\hat{\mathbf{z}}_{n,m} = (\hat{\mathbf{b}}_{n,m}, \hat{c}_{n,m})$ be the prediction pair. Second, we identify and group the detections corresponding to $\hat{\mathbf{z}}_{n,m}$ across the inference space that have the same predicted class and an overlap with its bounding box greater than a certain threshold. Specifically, we create a set $\mathcal{A}_{n,m}$ for each $\hat{\mathbf{z}}_{n,m}$. This set consists of all predictions $\hat{\mathbf{z}}_{k,l}$ such that $\hat{c}_{n,m} = \hat{c}_{k,l}$ and IoU between $\hat{\mathbf{b}}_{n,m}$ and $\hat{\mathbf{b}}_{k,l}$ is larger than some threshold $\gamma$ (set to 0.5 throughout experiments) [25]. Here $k \neq n$ and $l$ is an arbitrary detection in $k_{th}$ MC forward pass.

$$\mathcal{A}_{n,m} = \{\forall_{k \neq n} \cup (\hat{\mathbf{b}}_{k,l}, \hat{c}_{k,l}), \mid \text{IoU}(\hat{\mathbf{b}}_{n,m}, \hat{\mathbf{b}}_{k,l}) > \gamma , \ \hat{c}_{k,l} = \hat{c}_{n,m} \}. \tag{10}$$

Finally, $\mathcal{A}_{n,m}$ is utilized to estimate the uncertainty in $\hat{\mathbf{z}}_{n,m}$. We aim to capture uncertainty in predicted confidences and localization. So, we first estimate the variance in predicted (class) confidences, center x, center y, and aspect ratio of the bounding boxes predictions in $\mathcal{A}_{n,m}$ and later aggregate them to construct a joint measure of uncertainty. Let $\{\psi\}_{j=1}^{J}$ and $\{\Psi\}_{j=1}^{J}$ be the vectors ($J$ denotes the length) containing variances and means of predicted confidences, center-x, center-y, and aspect ratio of the predicted bounding boxes in $\mathcal{A}_{n,m}$, respectively. Let $\Psi_{agg}$ be the combined mean, computed as $\Psi_{agg} = \frac{1}{J} \sum_{j=1}^{J} \Psi_j$. Then, the combined variance representing a single, joint measure of uncertainty is computed as:

$$u_{n,m} = \frac{1}{J} \sum_{j=1}^{J} [\psi_j + (\Psi_j - \Psi_{agg})^2]. \tag{11}$$

**Uncertainty-guided soft pseudo-targets:** We leverage (joint) uncertainty to modulate the one-hot encoded pseudo-targets, formed according to (selected) pseudo-labels, to account for the entropy in object detections under target domain. Let $\text{H}_i^k$ be $i_{th}$ location in the class $k$ one-hot encoded channel map corresponding to a detected bounding box, with uncertainty $u_i^k$ and (pseudo) class label $k$. The uncertainty-guided soft pseudo-target is constructed as:

$$\widehat{\text{H}}_i^k = \begin{cases} \text{H}_i^k.(1 - u_i^k) & \text{if } \bar{s}_i^k \geq \kappa_1 \\ \text{H}_i^k.\bar{s}_i^k.(1 - u_i^k) & \text{if } \kappa_2 \leq \bar{s}_i^k < \kappa_1 \end{cases} \tag{12}$$

Where $\bar{s}_i^k = \frac{1}{T} \sum_j \hat{s}_i^j$ is the mean (class) confidence of the predicted bounding boxes in $\mathcal{A}_i$. $\kappa_1$ is threshold for ranking between highly confident and relatively less confident detections, and $\kappa_2$ is the confidence threshold below which there is no detection considered.

| Methods | Sim10k | | KITTI | | Cityscapes | |
|---|---|---|---|---|---|---|
| | **D-ECE** $\downarrow$ | **AP@0.5** $\uparrow$ | **D-ECE** $\downarrow$ | **AP@0.5** $\uparrow$ | **D-ECE** $\downarrow$ | **AP@0.5** $\uparrow$ |
| **In-domain Detections** | | | | | | |
| Single-stage | 16.1 | 79.2 | 15.1 | 94.1 | 15.1 | 44.9 |
| Single-stage + post-hoc | 23.4 | 78.7 | 22.8 | 94.1 | 25.3 | 44.3 |
| Single-stage + TCD | 14.9 | 83.4 | 12.6 | 94.7 | 9.4 | 48.3 |
| **Out-of-domain Detections** | | | | | | |
| | **Sim10K→CS** | | **KITTI→CS** | | **CS→CS-foggy** | **CS→BDD100K** |
| Single stage | 12.2 | 37.6 | 13.0 | 37.4 | 18.4    20.4 | 19.5    19.5 |
| Single-stage + post-hoc | 20.7 | 38.1 | 23.2 | 37.5 | 22.7    20.3 | 27.4    19.3 |
| Single-stage + TCD | 9.6 | 42.4 | 8.9 | 40.3 | 5.5    22.4 | 12.4    22.0 |

Table 1: Calibration performance and test accuracy with single-stage detector (FCOS [34]) trained with its application specific losses. Calibration with post-hoc temperature scaling and with our proposed TCD loss.

# 4 Experiments

**Datasets: Cityscapes** dataset [1] consists images of road and street scenes with bounding box annotations of following object categories: *person, rider, car, truck, bus, train, motorbike, and bicycle*. **KITTI** dataset [4] offers images of road scenes, captured from different viewing angle than Cityscapes and encompass wide-view of the area. **Foggy Cityscapes** dataset [31] is constructed by simulating foggy-weather, using depth maps, over the images in Cityscapes dataset. From the available three levels of fog, per the norm, we use the one with most dense fog. **Sim10k** dataset [12] is a collection of 10K images, synthesized from Grand Theft Auto V, and corresponding bounding box annotations of cars in those images. Follow prior works, while adapting from KITTI or Sim10k, only car-class is considered. Following prior works, car-class is **BDD100k** dataset [39] contains 100k annotated images with bounding boxes and category labels. 70k images are used for training and 30k are used for validation. Following [38] we only consider daylight images to create a training subset of 36.7k images and a validation subset of 5.2k images. Note that, the validation subset is used as an evaluation set.

**Evaluation and implementation details:** We report calibration performance using detection expected calibration error (D-ECE) and also report test accuracy. Further, we also plot reliability diagrams for visualizing calibration. Our TCD loss is developed to be used with the task-specific loss of DNN-based object detectors, including the SOTA domain-adaptive ones. For instance, the task specific losses for single-stage detectors use Focal loss [21] and IoU loss. Similarly Smooth L1 loss and Cross-Entropy are used in training the two-stage detectors. Let $\mathcal{L}_D$ be the task-specific loss, then the total loss in our method is computed as: $\mathcal{L} = \mathcal{L}_D + \mathcal{L}_{TCD}$. For further implementation details, refer to the supplementary.

**Considered object detectors:** Most current state-of-the-art (SOTA) object detectors are built on either a single-stage or a two-stage backbone. To include a representative detector models, from single-stage detectors we consider a state-of-the-art detector FCOS [34] and from multi-stage detectors we choose Faster-RCNN [29]. Our choice is partially motivated by the fact that the same backbones are also used in the current SOTA domain-adaptive object detectors. For the domain adaptive detectors we include EPM [9] and also SSAL [25] which are among the best performing single-stage domain adaptive detectors built on FCOS as source model. For the two-stage domain adaptive variant we include SWDA [30] that is built on FasterRCNN.

## 4.1 Experiments with One-stage and Two-stage Detectors

In this setting, we train a DNN-based object detector (1) with its task-specific loss functions and (2) after adding our TCD auxiliary loss function. For comparison we also include a post-hoc calibration technique based on temperature scaling. The temperature parameter $T$ is obtained using a hold-out validation set to perform temperature scaling at inference time in the target domain. We measure the performance and calibration errors in two settings. Here we measure the performance on the test set belonging to the same domain from which training dataset was sampled **In-domain Detection:** Here we measure the performance over the test-set partition of same dataset from which training set was extracted and **Out-of-domain Detection:** where we test it on an unseen target domain i.e., a different dataset.

| Methods | Sim10K | | CS | | | |
|---|---|---|---|---|---|---|
| | D-ECE | AP@0.5 | D-ECE | AP@0.5 | D-ECE | AP@0.5 |
| **In-domain Detections** | | | | | | |
| Two-stage | 23.7 | 66.2 | 16.2 | 38.3 | - | - |
| Two-stage + post-hoc | 15.0 | 66.3 | 13.6 | 38.0 | - | - |
| Two-stage + TCD | 14.5 | 66.7 | 11.0 | 39.9 | - | - |
| **Out-of-domain Detections** | | | | | | |
| | Sim10K → CS | | CS → CS-foggy | | CS → BDD100K | |
| Two-stage | 13.9 | 33.9 | 8.4 | 22.7 | 16.3 | 23.3 |
| Two-stage + post-hoc | 11.9 | 34.1 | 6.3 | 22.7 | 13.7 | 23.3 |
| Two-stage + TCD | 11.1 | 33.9 | 5.7 | 25.2 | 9.6 | 23.5 |

Table 2: Calibration results with a two-stage detector (Faster-RCNN [29]) trained with its task-specific loss, applying post-hoc temperature scaling on a pre-trained two-stage detector, and training two-stage detector after adding our TCD loss.

| Methods | Sim10K→CS | | CS→CS-foggy | |
|---|---|---|---|---|
| | D-ECE | AP@0.5 | D-ECE | AP@0.5 |
| **Single-stage Domain Adaptive Detectors** | | | | |
| EPM [9] | 16.0 | 46.7 | 15.7 | 38.6 |
| EPM + TCD | 9.9 | 47.7 | 14.8 | 39.9 |
| SSAL(UGPL) [25] | 13.6 | 49.5 | 22.1 | 35.0 |
| SSAL(UGPL) + TCD | 8.5 | 51.4 | 19.1 | 35.2 |
| **Two-stage Domain Adaptive Detectors** | | | | |
| | Sim10K→CS | | CS→CS-foggy | |
| SWDA [30] | 14.6 | 40.9 | 11.2 | 36.6 |
| SWDA + TCD | 13.8 | 40.0 | 9.3 | 37.5 |

Table 3: Calibration results with two different single-stage domain-adaptive detectors, EPM [9] and SSAL(UGPL) [25], and a two-stage detector SWDA [30]. We report both calibration performance (ECE) and test accuracy (AP@0.5).

Table 1 reports results with a one-stage detector (FCOS [34]) trained with its task-specific losses i.e. Focal loss and IoU loss. We report calibration errors (D-ECE) and performance of the model. We show the impact on calibration after adding and training with the proposed auxiliary loss TCD. For comparison we include post-hoc temperature scaling based calibration method. For in-domain detections, we see that our proposal, single-stage detector trained after adding TCD, achieves the best D-ECE score across all datasets. Furthermore, our proposal allows boosting the detection performance by notable margins compared to a single-stage detector in all datasets. For out-of-domain detections, we significantly improve the calibration performance over baselines in all domain shift scenarios. For instance, it decreases the D-ECE score by 12.9% and 7.1% compared to the single-stage detector in CS→CS-foggy and CS→BDD100K, respectively. At the same time, it provides notable gains in detection performance over single-stage detector e.g., a 4.8% gain (AP@0.5) in Sim10K→CS.

We report results with a two-stage detector Faster-RCNN [29], trained with a task-specific loss, taking a pre-trained two-stage detector and applying post-hoc temperature scaling, and a two-stage trained with task-specific loss and our TCD loss (Table 2). For both in-domain and out-of-domain detections, we observe that our proposal, two-stage detector trained after adding TCD, delivers the best D-ECE score across all datasets.

## 4.2 Experiments with Domain-adaptive Detectors

In this setting, we incorporate our TCD auxiliary loss in the recent SOTA domain-adaptive detectors and observe impact on calibration performance. In single-stage paradigm, we chose EPM [9], which accounts for pixel-wise centerness and objectness for target domain images in an adversarial alignment framework. Further, we also choose a method from the line of self-training based domain-adaptive detectors. Particularly, we select uncertainty-guided pseudo-labelling (UGPL) from SSAL [25]. In two-stage paradigm, we choose SWDA [30], which proposes strong local alignment and weak global alignment in an adversarial alignment framework.

Table 3 reports results with two single-stage domain-adaptive detectors: EPM [9], UGPL from [25]. After adding our proposed loss TCD, the calibration performance for both domain-adaptive detectors is significantly improved. In Sim10K→CS, both EPM+TCD and SSAL(UGPL)+TCD reduce ECE score by 6.1% and 5.1%, respectively. Further, we observe that after adding our auxiliary loss (TCD), there are notable gains in AP@0.5.

We show the impact of our TCD loss on the calibration performance of a two-stage domain-adaptive detector (SWDA [30]) in Table 3. Upon adding our loss TCD in SWDA, we observe that the calibration of SWDA is improved in both adaptation scenarios.

## 4.3 Experiments with Implicit Calibration Technique

We validate the effectiveness of our implicit calibration technique (ICT) (sec. 3.3) by integrating in a self-training based uncertainty-guided pseudo-labelling baseline (UGPL)) from [25]. UGPL is a strong baseline in terms of benchmarking model calibration because it leverages uncertainty

| Method/Shift scenarios | Sim10k → CS | | CS → CS-foggy | |
|---|---|---|---|---|
| | D-ECE | AP@0.5 | D-ECE | mAP@0.5 |
| SSAL(UGPL) [25] | 13.6 | 49.5 | 22.1 | 35.0 |
| SSAL(UGPL)+ICT | 12.7 | 51.3 | 19.5 | 34.2 |
| SSAL(UGPL)+TCD | 8.5 | 51.4 | 19.1 | 35.2 |
| SSAL(UGPL)+ICT+TCD | 7.9 | 50.7 | 16.7 | 36.9 |

Table 4: Calibration results with our implicit calibration technique (ICT).

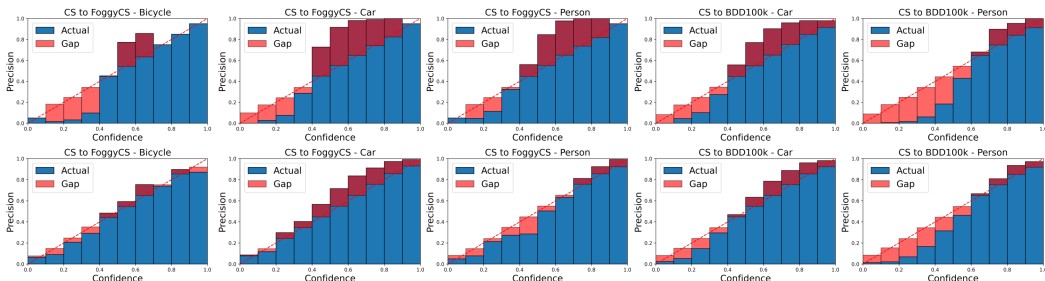

Figure 2: Class-wise reliability diagrams. Top row: One-stage detector trained with task-specific loss. Bottom row: One-stage detector trained with task-specific loss and our loss TCD.

to select pseudo instance-level labels in an unlabelled target domain for self-training. Table 4 reports results for following methods: SSAL(UGPL) [25], SSAL(UGPL)+ICT, SSAL(UGPL)+TCD, SSAL(UGPL)+ICT+TCD. We see that SSAL(UGPL)+ICT decreases D-ECE by 0.9% in Sim10→CS shift. Further, adding TCD shows significant improvement in calibration performance (D-ECE). Finally, using both ICT and TCD achieves the best calibration scores in both Sim10K→CS and CS→CS-foggy shifts, thereby revealing that ICT and TCD are complementary.

## 4.4 Ablation Study and Analysis

For ablation experiments with TCD, we use one-stage detector (FCOS [34]. For ICT ablation experiments, we use the UGPL component of SSAL [25], which is a domain-adaptive detector.

**Mitigating Under/Over-Confidence:** Fig. 2 shows that our proposal, task-specific loss in conjunction with proposed TCD loss, facilitates a one-stage detector in mitigating both under-confident and over-confident detections in two different multi-class shift scenarios.

**Impact on test accuracy:** We observe the impact on test accuracy after adding our TCD loss with the task-specific loss of one-stage detector in Table 5. Our proposal is capable of providing gains in test accuracy with visible margins over higher IoU thresholds and over the spectrum of large, medium and small objects.

| Shift scenario | Method | AP(mean) | AP@0.5 | AP@0.75 | AP@S | AP@M | AP@L |
|---|---|---|---|---|---|---|---|
| Sim10K → CS | One-stage | 17.8 | 37.6 | 14.9 | 3.8 | 19.7 | 36.6 |
| | One-stage+TCD | 22.5 | 42.4 | 21.4 | 4.2 | 24.8 | 46.5 |
| CS → BDD100K | One-stage | 9.5 | 19.5 | 7.9 | 3.5 | 11.7 | 20.0 |
| | One-stage+TCD | 10.7 | 22.0 | 9.0 | 3.2 | 13.2 | 23.7 |

Table 5: Test accuracy after adding our TCD loss with the task-specific loss of one-stage detector (FCOS [34]).

**Without detection component** $d_{det}$**:** Table 6 shows the impact on calibration performance without $d_{det}$ component in TCD. We see that without $d_{det}$, the calibration performance (D-ECE) degrades.

| Scenarios | Sim10k to CS | | | | CS to CS-foggy | | | | KITTI to CS | | | | CS to BDD100K | | | |
|---|---|---|---|---|---|---|---|---|---|---|---|---|---|---|---|---|
| | OOD | | In-domain | | OOD | | In-domain | | OOD | | In-domain | | OOD | | In-domain | |
| | D-ECE | AP@0.5 | D-ECE | AP@0.5 | D-ECE | AP@0.5 | D-ECE | AP@0.5 | D-ECE | AP@0.5 | D-ECE | AP@0.5 | D-ECE | AP@0.5 | D-ECE | AP@0.5 |
| w/o d_det | 10.3 | 44.9 | 15.2 | 82.3 | 8.1 | 23.8 | 13.3 | 44.8 | 11.4 | 38.7 | 13.0 | 94.3 | 16.5 | 19.8 | 13.3 | 44.8 |
| with d_det | 9.6 | 42.4 | 14.9 | 83.4 | 5.5 | 22.4 | 9.4 | 48.3 | 8.9 | 40.3 | 12.6 | 94.7 | 12.4 | 22.0 | 9.4 | 48.3 |

Table 6: Impact on calibration performance without $d_{det}$ component of TCD in four domain shift scenarios.

**On different uncertainty quantification methods for ICT:** Table 7 reports the calibration performance of ICT with different methods of quantifying uncertainty. We use variances across: predicted confidences only [25] $u_{conf.}$; predicted confidences and center-x,center-y $u_{conf.,x,y}$; predicted confidences and center-x,center-y, aspect-ratio (ours) $u_{conf.,x,y,ar}$. Among different methods, our proposed technique of estimating detection uncertainty in ICT provides the lowest D-ECE score.

| Methods (Sim10K→CS) | D-ECE | AP@0.5 |
|---|---|---|
| SSAL(UGPL) [25] | 13.6 | 49.5 |
| SSAL(UGPL)+ICT($u_{conf.}$) | 13.0 | 50.5 |
| SSAL(UGPL)+ICT($u_{conf.,x,y}$) | 12.8 | 51.1 |
| SSAL(UGPL)+ICT($u_{conf.,x,y,ar}$) | 12.7 | 51.3 |

Table 7: Calibration performance of ICT when using different ways of quantifying detection uncertainty. See text for details.

**Limitation:** Current study does not explore the impact on calibration with respect to objects sizes. Further, our train-time calibration loss could potentially result in less improvement in classes with relatively less instances.

## 5 Conclusion

In this paper, we approached the challenging problem of calibrating DNN-based object-detectors for in-domain and out-of-domain detections and improving their calibration in the domain-adaptation context. We proposed an auxiliary train-time calibration loss TCD that jointly calibrates the class-wise confidences and localization performance. Further, we develop an implicit calibration technique (ICT) for self-training based domain adaptive detectors. Results show that TCD loss is capable of improving calibration of both one-stage and two-stage detectors. Finally, we show that ICT and TCD together results in well-calibrated domain-adaptive detectors.

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
