# Supplementary Material: Towards Improving Calibration in Object Detection Under Domain Shift

**Muhammad Akhtar Munir**[1,2] **Muhammad Haris Khan**[2]**, M. Saquib Sarfraz**[3,4]**, Mohsen Ali**[1]

[1] Information Technology University of Punjab, [2] Mohamed bin Zayed University of Artificial Intelligence,
[3] Karlsruhe Institute of Technology, [4]Mercedes-Benz Tech Innovation

In this supplementary material, we first present the following (additional) results of our calibration techniques with: another calibration metric, a recent transformer-based object detector, and a recent domain-adaptive detector. Next, we report the calibration performance of our proposed loss (TCD) without the classification component $d_{cls}$. We also present results on large scale datasets (e.g. MS-COCO and PASCAL-VOC). Finally, we show some qualitative results of our proposed train-time calibration loss and describe the implementation details for different detectors considered.

## 1 Results with Detection Expected Uncertainty Calibration Error (D-UCE)

For detectors that leverage uncertainty, in addition to (D-ECE), we also report detection expected uncertainty calibration error (D-UCE) [3]:

$$ D - UCE = \sum_{m=1}^{M} \frac{|I(m)|}{|\mathcal{D}|} \left| error(m) - uncertainty(m) \right| . \tag{1} $$

Where $error(m)$ denotes the average error in a bin and $uncertainty(m)$ represents the average uncertainty in a bin. $I(m)$ is the set of samples in $m^{th}$ bin, and $|\mathcal{D}|$ is the total number of samples. The error for a particular sample (detection) is computed as: $\mathbb{1}[IoU(\hat{\mathbf{b}}_m, \mathbf{b}_m^*) < 0.5]\mathbb{1}[\hat{c}_m \neq c_m^* \vee \hat{c}_m = c_m^*]$.

Table 1 reports calibration performances of SSAL(UGPL), SSAL(UGPL)+ICT, SSAL(UGPL)+TCD, SSAL(UGPL)+ICT+TCD in D-UCE and D-ECE metrics. We see that our calibration techniques, when either used individually or as a combination, can not only decrease the D-ECE but are also capable of reducing the D-UCE.

| Method/Shift scenarios | Sim10k $\rightarrow$ CS | | | CS $\rightarrow$ CS-foggy | | |
|---|---|---|---|---|---|---|
| | D-ECE | D-UCE | AP@0.5 | D-ECE | D-UCE | mAP@0.5 |
| SSAL(UGPL) [4] | 13.6 | 15.9 | 49.5 | 22.1 | 26.2 | 35.0 |
| SSAL(UGPL)+ICT | 12.7 | 14.8 | 51.3 | 19.5 | 25.0 | 34.2 |
| SSAL(UGPL)+TCD | 8.5 | 10.0 | 51.4 | 19.1 | 24.3 | 35.2 |
| SSAL(UGPL)+ICT+TCD | 7.9 | 11.5 | 50.7 | 16.7 | 21.7 | 36.9 |

Table 1: Calibration performance in terms of detection expected uncertainty calibration error (D-UCE). We show calibration performances of SSAL(UGPL), SSAL(UGPL)+ICT, SSAL(UGPL)+TCD, SSAL(UGPL)+ICT+TCD.

## 2 Experiments with Transformer-based Object Detector

In addition to one-stage and two-stage object detectors, we also reveal the effectiveness of our train-time calibration loss (TCD) towards calibrating recent transformer-based object detectors. Particularly, we chose the Deformable Detr object detector [10] and integrate our TCD loss. Table 2 shows that our TCD loss improves the calibration of Deformable Detr detector for both in-domain and

out-of-domain detections. However, the calibration performance is more pronounced for in-domain detections as compared to the out-of-domain detections.

| Methods/Scenarios | Sim10k to CS | | | | CS to Foggy | | | |
| --- | --- | --- | --- | --- | --- | --- | --- | --- |
| | OOD | | InDomain | | OOD | | InDomain | |
| | D-ECE | AP@0.5 | D-ECE | AP@0.5 | D-ECE | AP@0.5 | D-ECE | AP@0.5 |
| Deformable-Detr | 9.0 | 48.0 | 14.9 | 90.9 | 8.2 | 29.5 | 16.1 | 48.3 |
| Deformable-Detr + post-hoc | 10.9 | 48.0 | 7.8 | 90.9 | 13.4 | 29.5 | 17.5 | 48.3 |
| Deformable-Detr + TCD | 7.5 | 48.4 | 6.1 | 90.7 | 7.9 | 30.2 | 15.7 | 46.0 |

Table 2: Calibration results with Deformable Detr [10] trained with its task-specific loss, applying post-hoc temperature scaling on a pre-trained Deformable Detr, and training Deformable Detr after adding our TCD loss.

## 3 Without $d_{cls}$ component in TCD

Table 3 reports the impact on calibration performance upon excluding the $d_{cls}$ component of TCD. We observe a significant drop in calibration performance without $d_{cls}$ component. A similar drop in calibration performance can be seen without $d_{det}$ component. We empirically show that both components are complementary and so are vital for the effectiveness of TCD loss.

| Scenarios | Sim10k to CS | | | | CS to CS-foggy | | | | KITTI to CS | | | | CS to BDD100K | | | |
| --- | --- | --- | --- | --- | --- | --- | --- | --- | --- | --- | --- | --- | --- | --- | --- | --- |
| | OOD | | In-domain | | OOD | | In-domain | | OOD | | In-domain | | OOD | | In-domain | |
| | D-ECE | AP@0.5 | D-ECE | AP@0.5 | D-ECE | AP@0.5 | D-ECE | AP@0.5 | D-ECE | AP@0.5 | D-ECE | AP@0.5 | D-ECE | AP@0.5 | D-ECE | AP@0.5 |
| w/o d_det | 10.3 | 44.9 | 15.2 | 82.3 | 8.1 | 23.8 | 13.3 | 44.8 | 11.4 | 38.7 | 13.0 | 94.3 | 16.5 | 19.8 | 13.3 | 44.8 |
| w/o d_cls | 10.2 | 38.0 | 15.5 | 79.9 | 13.3 | 23.6 | 14.6 | 44.6 | 9.1 | 38.6 | 11.2 | 95.0 | 18.5 | 21.1 | 14.6 | 44.6 |
| TCD | 9.6 | 42.4 | 14.9 | 83.4 | 5.5 | 22.4 | 9.4 | 48.3 | 8.9 | 40.3 | 12.6 | 94.7 | 12.4 | 22.0 | 9.4 | 48.3 |

Table 3: Impact on calibration performance without $d_{cls}$ component of TCD in four domain shift scenarios.

## 4 Performance of our loss with SSAL (UGPL+UGT)

Table 4 reports calibration performance with domain-adaptive detector SSAL(UGPL+UGT) [4] and SSAL(UGPL+UGT) with TCD. We see that SSAL(UGPL+UGT) with TCD significantly improves the calibration performance of SSAL(UGPL+UGT).

## 5 Results on COCO

We include our results on several datasets that are commonly used to study detection performance under domain shift. Results on COCO would also be interesting to see the impact on calibration performance. We, therefore, provide results with our TCD loss and related ablation analysis on the COCO dataset below in Table 5, Table 6, and Table 7. For Pascal VOC, please also see Table 8. For in-domain COCO results, we evaluate our trained model(s) on COCO2017 minival (5K) dataset (Table 5). For out-of-domain COCO results, we evaluate our trained model(s) on two different out-of-domain scenarios. These are curated by systematically corrupting the COCO minival set images. The corrupted versions are obtained by following the proposals in [1, 9]. The first OOD scenario is produced by adding a fixed corruption (fog) and fixed severity. See Table 6 (top) for results. Whereas the second is generated by first randomly choosing a corruption (out of 19 different corruption modes) and then randomly sampling their severity level (from 1-5) (see Table 7). We note that our proposed TCD loss is capable of improving the calibration of both in-domain and out-of-domain detections. Further, both the detection ($d_{det}$) and the classification ($d_{cls}$) components of our TCD loss are integral towards boosting the calibration performance.

## 6 Implementation Details

Note that, for every individual method considered in our experiments, we use its default training and testing specifics. All experiments are performed using a single GPU (Quadro RTX 6000).

**One-stage detector**: FCOS [8] is a one-stage anchor-less object detector. To integrate our TCD loss with FCOS, we simple add our TCD loss with the task-specific loss of FCOS, which itself comprises

| Method/Scenarios | Sim10k to CS | | |
|---|---|---|---|
| | D-ECE | AP (mean) | AP@0.5 |
| SSAL (UGPL + UGT) | 14.1 | 28.9 | 51.8 |
| SSAL (UGPL + UGT) with TCD | 11.3 | 29.7 | 51.6 |

Table 4: Calibration results with SSAL(UGPL+UGT) [4] and SSAL(UGPL+UGT) with our TCD loss. We report both calibration performance (D-ECE) and test accuracy (AP@0.5 and AP(mean)).

| In Domain COCO | | |
|---|---|---|
| | D-ECE | mAP@0.5 |
| Single Stage | 24.0 | 50.5 |
| Single Stage + TCD (w/o d_cls) | 23.3 | 50.2 |
| Single Stage + TCD (w/o d_det) | 23.4 | 50.4 |
| Single Stage + TCD | 23.3 | 50.8 |

Table 5: Ablation studies on in-domain COCO dataset.

of focal loss and IoU loss, to get a joint loss which is minimized to achieve train-time calibration with our TCD loss.

For EPM [2], we use source ground-truth labels for our TCD loss and add it to task-specific losses of EPM to obtain a joint loss which is optimized during adaptation. However, for SSAL [4], we utilize both source ground-truth and target pseudo-labels to create two instances of our TCD loss which are then added to SSAL task-specific losses to obtain a joint loss.

We integrate our ICT component in the UGPL module of SSAL [4]. Specifically, the selected pseudo-labels from UGPL module are converted to soft pseudo-targets, using Eq.(10-12), to be used in task-specific loss. The values of $\kappa_1$ and $\kappa_2$ in Eq.(12) of main paper are set to 0.75 and 0.5, respectively, in all experiments.

**Two Stage Detector**: For both Faster RCNN [5] and SWDA [7], we utilize the output of second stage (i.e. Fast RCNN module) to implement our TCD loss and combine it with the respective task-specific losses to obtain a joint loss formulation which is then optimized for training.

**Deformable Detr**: We add our TCD loss with the task-specific losses (focal loss and generalized IoU loss [6]) of Deformable Detr [10] to acquire a joint loss which is then optimized during training.

## 7 Qualitative Results

Fig. 1 visualizes some calibration results for out-of-domain detections with one-stage detector and one-stage detector with our TCD loss. We see that one-stage detector trained with our TCD loss facilitates improved localization performance and increased confidence score per detected instance of an object. Furthermore, it allows a decrease in confidence score for wrong detections.

| OUT Domain COCO (fix corrupted) | | |
|---|---|---|
| | D-ECE | mAP@0.5 |
| **Single Stage** | 22.2 | 37.6 |
| **Single Stage + TCD (w/o d_cls)** | 21.2 | 37.8 |
| **Single Stage + TCD (w/o d_cet)** | 21.4 | 38.2 |
| **Single Stage + TCD** | 20.9 | 38.1 |

Table 6: Ablation studies on two out-of-domain COCO scenarios. The out-of-domain images with a fixed corruption (fog) and fixed severity.

| OUT Domain COCO (random corrupted) | | |
|---|---|---|
| | D-ECE | mAP@0.5 |
| **Single Stage** | 23.3 | 27.9 |
| **Single Stage + TCD (w/o d_cls)** | 22.5 | 28.1 |
| **Single Stage + TCD (w/o d_det)** | 22.7 | 28.3 |
| **Single Stage + TCD** | 22.4 | 28.1 |

Table 7: Ablation studies on two out-of-domain COCO scenarios. Randomly choosing the corruption and then randomly sampling its severity level.

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

| COCO to VOC | Single Stage | D-ECE | AP (mean) | mAP@0.5 |
|---|---|---|---|---|
| | Baseline | 26.1 | 47.9 | 72.0 |
| | TCD | 25.5 | 48.2 | 72.0 |

Table 8: Calibration and detection performance upon training a model on COCO and testing it on PASCAL VOC 2012.

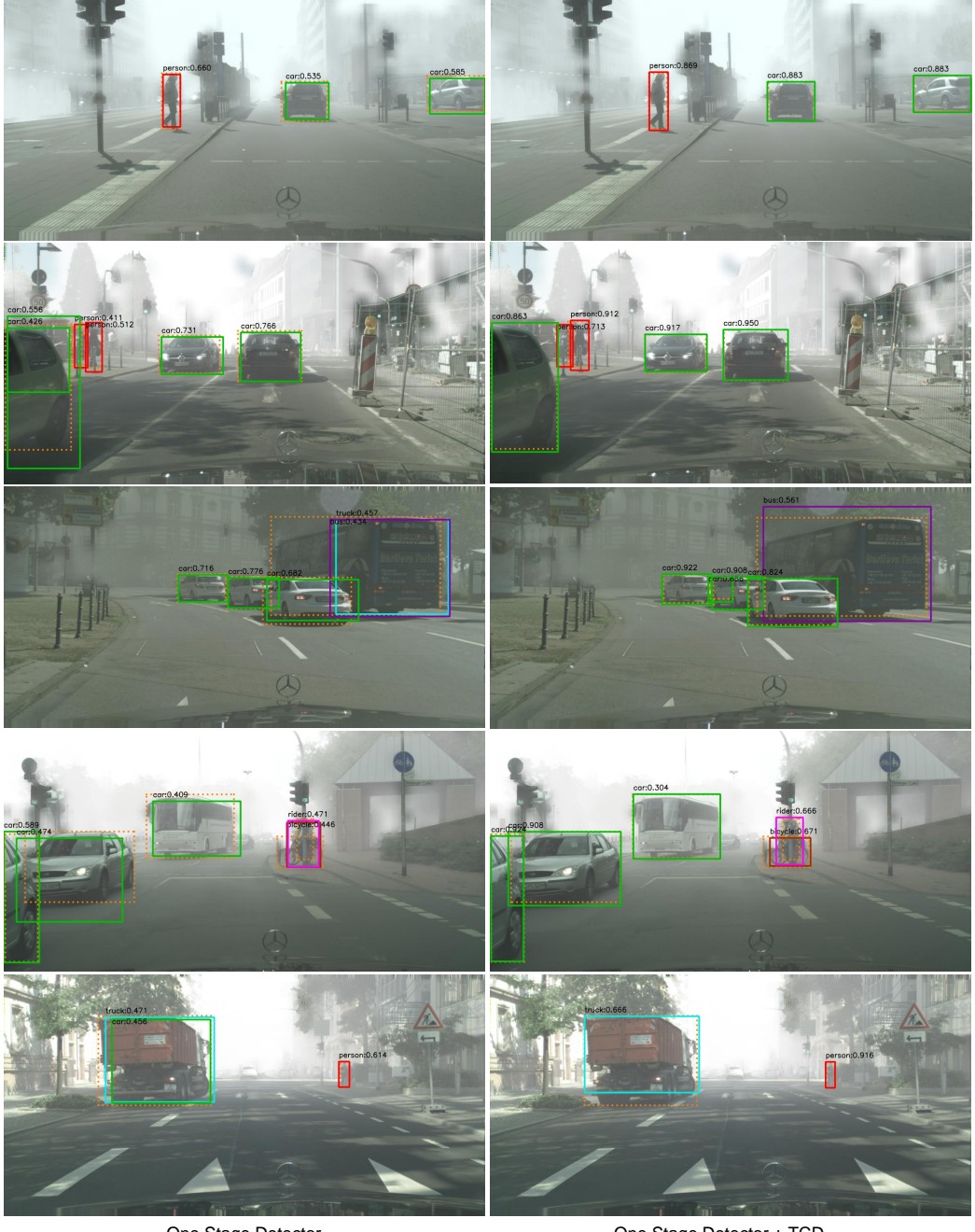

One Stage Detector                    One Stage Detector + TCD

Figure 1: Visual depiction of calibration results for out-of-domain detections with one-stage detector (left column) and one-stage detector trained with our TCD loss (right column). Dotted bounding boxes are the ground truth and solid bounding boxes are the detections. We use a distinct colored bounding box for each object category. To avoid clutter, we only draw ground truth bounding boxes corresponding to detections. Best viewed in color and zoom.