# OpenReview forum: "Towards Improving Calibration in Object Detection Under Domain Shift"
_NeurIPS.cc/2022/Conference — NeurIPS 2022 Accept_

### Official Review · Reviewer_B3RR · 2022-07-11

**Rating:** 5
**Confidence:** 3
**Soundness:** 3 good
**Presentation:** 2 fair
**Contribution:** 3 good

**Summary:**

This works identifies the calibration in object detection under domain shift. The author proposes a new plug-and-play loss formulation, termed as train-time calibration for detection (TCD). It can be used in some task-specific loss functions during training phase and acts as a regularization for detections.


**Questions:**

- Ablation study of the work in some well-known datasets like COCO and VOC.

**Limitations:**

- Ablation study of the work in some well-known datasets like COCO and VOC.
- Not very practical for general object detection, although it has some theoretical analysis.

**Strengths And Weaknesses:**

- The author uses mathematical deduction to analyze the domain shift in detection. The Calibration including TCD, Quantifying and uncertainty-guided soft pseudo-targets can easily be plugged into many detection frameworks.

---

> ### Author Response · Authors · 2022-08-02
> **Response to Reviewer B3RR**
>
> Thank you for your positive feedback.
>
> Q1: "Ablation study of the work in some well-known datasets like COCO and VOC"
>
> A1: We include our results on several datasets that are commonly used to study detection performance under domain shift. As pointed out, we agree that results on COCO would also be interesting to see the impact on calibration performance. We, therefore,  provide results with our TCD loss and related ablation analysis on the COCO dataset below in Table-C & Table-D.  For Pascal VOC please also see our response for Reviewer ETNZ Table B. We will include these results in our supp.
>
> For in-domain COCO results, we evaluate our trained model(s) on COCO2017 minival (5K) dataset (Table-C).
>
> For out-of-domain COCO results, we evaluate our trained model(s) on two different out-of-domain scenarios.  These are curated by systematically corrupting the COCO minival set images. The corrupted versions are obtained by following the proposals in  [a,b] (see these references below).
>
> The first OOD scenario is produced by adding a fixed corruption (fog) and fixed severity. See Table-D (top) for results. Whereas the second is generated by first randomly choosing a corruption (out of 19 different corruption modes) and then randomly sampling their severity level (from 1-5) (see Table-D (bottom)). We note that our proposed TCD loss is capable of improving the calibration of both in-domain and out-of-domain detections. Further, both the detection (d_det) and the classification (d_cls) components of our TCD loss are integral towards boosting the calibration performance.
>
>
> | In   Domain COCO                     |      |         |
> |--------------------------------------|------|---------|
> |                                      | D-ECE | mAP@0.5 |
> | Single   Stage                       | 24.0   | 50.5    |
> | Single   Stage + TCD (w/o d_cls)      | 23.3 | 50.2    |
> | Single   Stage + TCD (w/o d_det)        | 23.4 | 50.4    |
> | Single   Stage + TCD                 | 23.3 | 50.8    |
>
>
> Table-C: Ablation studies on in-domain COCO dataset.
>
> | OUT   Domain COCO (fix corrupted)    |      |         |
> |--------------------------------------|------|---------|
> |                                      | D-ECE | mAP@0.5 |
> | Single   Stage                       | 22.2 | 37.6    |
> | Single   Stage + TCD (w/o d_cls)      | 21.2 | 37.8    |
> | Single   Stage + TCD (w/o d_cet)        | 21.4 | 38.2    |
> | Single   Stage + TCD                 | 20.9 | 38.1    |
>
>
> | OUT   Domain COCO (random corrupted) |      |         |
> |--------------------------------------|------|---------|
> |                                      | D-ECE | mAP@0.5 |
> | Single   Stage                       | 23.3 | 27.9    |
> | Single   Stage + TCD (w/o d_cls)      | 22.5 | 28.1    |
> | Single   Stage + TCD (w/o d_det)        | 22.7 | 28.3    |
> | Single   Stage + TCD                 | 22.4 | 28.1    |
>
> Table D: Ablation studies on two out-of-domain COCO scenarios. Top: The out-of-domain images with a fixed corruption (fog) and fixed severity. Bottom: randomly choosing the corruption and then randomly sampling its severity level. (see ref [a, b] for details)
>
>
> [a] Dan Hendrycks and Thomas Dietterich. Benchmarking neural network robustness to common corruptions and perturbations. In International Conference on Learning Representations (ICLR), 2019
>
> [b] Wang, Haotao, et al. "Augmax: Adversarial composition of random augmentations for robust training." Advances in neural information processing systems 34 (2021): 237-250.

---

> ### Author Response · Authors · 2022-08-09
> **Note to Reviewer B3RR**
>
> In response to the reviewer B3RR's highlighted single question about ablation study on COCO, we have provided the mentioned additional results.
>
> Since no further response was given during the discussion period, we would like to thank the reviewer once again and sincerely hope that they will consider our response before finalizing their scores.

---

### Official Review · Reviewer_ETNZ · 2022-07-11

**Rating:** 6
**Confidence:** 4
**Soundness:** 3 good
**Presentation:** 4 excellent
**Contribution:** 4 excellent

**Summary:**

The paper proposes a new well-calibrated detection algorithm for both in-domain and out-of-domain scenarios. The author brought the concepts from calibration for classification into detection, and proposed a calibration loss (TCD) for object detection by considering both classvcaition and localisation predictions. In addition, a uncertainty quantification mechanism is proposed to further boost the performance. The proposed method shows clear improvement over the baselines, and the experiments are comprehensive.

**Questions:**

Please refer "Strengths And Weaknesses"

**Limitations:**

I do not see any potential negative societal impact of their work.

**Strengths And Weaknesses:**

Pros:
- The idea of using the concepts of calibration for detection is important and novel to me.
- The paper is easy to understand, and the experiments are very comprehensive.
- The results are competitive and show clear improvements over baselines.

Cons:
- For Table 4 R4-6 C2-3, I feel a bit strange that both TCD and ICT can reduce the D-ECE score, however, when combining them together, the overall AP reduces with even lower D-ECE value. The similar results also appear in Table 2 R4-6 C8-9 (Supp), where the TCD variant is even worse than the baselines under AP metric, while still it achieves lower D-ECE score.
- Again For Table 2 (Supp), the results seem not promising when applying the proposed module in transformer-based detectors. The generalisation of the proposed modules becomes questionable.
- I learn the method can improve both in-domain and out-of-domain performance of object detection, so I wonder can we apply the proposed module into the existing SOTA detectors to reach a higher performance.
- Though it's commonly used in ood scenario, I am curious the performance of applying COCO trained detector into VOC datasets directly.

Overall speaking, I think it is a good paper.

---

> ### Author Response · Authors · 2022-08-02
> **Response to Reviewer ETNZ**
>
> Thank you for finding our work to be important and novel. We appreciate the input and questions.
>
> Q1: For Table 4 R4-6 C2-3, I feel a bit strange that both TCD and ICT […]
>
> A1: The scope of the proposed work lies around improving the calibration of modern DNN-based object detectors and their adaptation. In almost all results, our proposed methods improve calibration and show a notable decrease in D-ECE, including when applied to domain adaptive detectors. While not explicitly targeting average precision improvement we observe (notable) gains in the AP@0.5 except in these two instances where we see a slight decrease under AP metric.
>
>
> ***
>
>
> Q2: Again For Table 2 (Supp), the results seem not promising when [...]
>
> A2:  Transformer-based detector is a much stronger baseline, in terms of calibration performance than the single-stage/two-stage detectors.  As such, the margin for improving calibration performance in deformable-detr seems rather small in comparison. Nonetheless, Table 2 shows an improvement for D-ECE on this transformer-based detector as well.
>
> ***
>
> Q3: "Though it's commonly used in ood scenario, I am curious the performance of applying COCO trained detector into VOC datasets directly."
>
> A3: As suggested, we trained the baseline (single-stage detector) and our approach (single-stage detector + TCD) on COCO dataset and evaluated them on the val. set of PASCAL VOC 2012 dataset (see Table-B). We observe that, compared to baseline, our approach improves calibration performance as well as the average precision accuracy AP(mean).
>
> | COCO   to VOC | Single Stage | D-ECE | AP (mean) | mAP@0.5 |
> |---------------          |--------------      |------     |-----------      |-----|
> |                            | baseline         | 26.1    | 47.9          | 72.0 |
> |                 	      | TCD               | 25.5    | 48.2           | 72.0  |
>
>
> Table-B: Calibration and detection performance upon training a model on COCO and testing it on PASCAL VOC 2012.

---

> > ### Comment · Reviewer_ETNZ · 2022-08-06
> > **Final Rating**
> >
> > My concerns have been resolved by the rebuttal. I will keep the original score.

---

### Official Review · Reviewer_kwna · 2022-07-11

**Rating:** 8
**Confidence:** 4
**Soundness:** 3 good
**Presentation:** 3 good
**Contribution:** 3 good

**Summary:**

In this paper, the authors propose two methods to improve the calibration of confidence scores of object detection models based on deep neural networks. A first method, TCD, introduces a specific cost function that aims to constrain the model to reduce the difference between the predicted confidence and the prediction accuracy (d_cls) and to reduce the difference between the confidence and the IoU of the detected regions compared to the ground truth (d_det).
This cost function is used for supervised training of the models. A second method, ICT, aims to calibrate the predictions on a set of out-of-domain tests in an unsupervised (or self-supervised) manner. This method uses a MC-Dropout to generate multiple hypotheses and calculates a combined variance on the confidence, position and aspect ratio of predictions of the same classes among the hypotheses. This combined variance is then used to generate pseudo-targets for self-supervised adaptation.

These methods are evaluated on 5 object detection databases, using different detection models (FCOS, Faster-RCNN, EPM, SSAL, SWDA). The experiments show that the proposed methods improve both the detection expected calibration error (D-ECE) and the accuracy of the models, on all the bases and for all the models tested. An ablation study completes the evaluation.


**Questions:**

* Fig1: explain what is dark red and light red
* L125 avoid foot note in equation, could be read as exponent 1[IoU(bˆ, b∗) ≥ γ2]1[yˆ = y∗].
* Eq 7: sl,r[k]  is not defined
* L196: J is not defined

**Strengths And Weaknesses:**

The paper is well written, with clear description of the methods and very thorough experiments. The results are convincing. It is shown that the proposed methods can be applied to different types of models and on different databases.

However, the article has a small weakness in the comparison with other calibration methods: only the simple temperature scaling method is used as a basis for comparison. More complex methods could have been compared.  However, given the space constraints, it seems difficult to hold this against the authors.

---

> ### Author Response · Authors · 2022-08-02
> **Response to Reviewer kwna**
>
> Thank you for the positive comments and feedback. Below we clarify the points raised. We will also update them in the final draft.
>
> Q1: Fig1: explain what is dark red and light red
>
> A1: For a bin, both shades of red show the gap to perfect calibration, however, the dark red color represents under-confidence in detections whereas the light red denotes over-confidence in detections.
>
> ***
> Q2: L125 avoid footnote in equation [...].
>
> A2:  Thank you for highlighting this, we will update the draft.
> ***
>
> Q3: Eq 7: sl,r[k] is not defined.
>
> A3:  It is the k_th class confidence for r_th location in the output classification channel map. We will update the draft to define this.
> ***
>
> Q4: L196: J is not defined
>
> A4: Thank you for highlighting this. J denotes the length of vector 𝜓.
> ***
>
> Q5. “ comparison with other calibration methods “ [...]
>
> A5. Most other train-time calibration methods have been proposed in the context of classification problems. However, for completeness and as suggested, we compare our method (TCD) with another such calibration loss namely DCA from [21] and include the results below (see Table-A).  The compared calibration loss (DCA [21]) also acts as an auxiliary loss as it strives to minimize the difference between accuracy and the confidence of the predicted class.  We note that our proposed TCD loss delivers better calibration results than the DCA [21] in two different domain-shift scenarios. Further, it also provides an improved gain in mAP compared to DCA [21] in both shift scenarios.
>
>
> |                	 | CS to foggy CS |                          | CS to BDD100k   |      |
> |----------------    |----------------        |------------     |---------------            |------|
> | Single   Stage | D-ECE                | mAP@0.5         | D-ECE                | mAP@0.5  |
> | baseline       	 | 16.8                    | 20.4         | 15.7                   | 19.5 |
> | DCA [21]         | 5.2                     | 21.7         | 11.1        	           | 20.4 |
> | TCD (ours)     | 4.0                     | 22.4           | 9.6                      | 22.0 |
>
>
>
> Table-A: Comparison with another train-time calibration loss, namely DCA from [21].

---

### Official Review · Reviewer_GHp5 · 2022-07-12

**Rating:** 6
**Confidence:** 3
**Soundness:** 3 good
**Presentation:** 3 good
**Contribution:** 3 good

**Summary:**

The paper focuses on the calibration of both two-stage and one-stage object detectors. Moreover, the author conducts studies on in-/out-domain distribution.
Specifically, an auxiliary loss function (TCD), which is based on the distance between class accuracy and bounding box IOU, is introduced.
In addition, a quantification mechanism is proposed for self-training of the detectors.

**Questions:**

see Weaknesses

**Limitations:**

see Weaknesses

**Strengths And Weaknesses:**

[Strengths]
1. The proposed method is simple and reasonable.
2. The paper is clearly presented and well-organized.
3. The proposed method outperforms all baselines.

[Weaknesses]
In Table 1 and 2, it raises my attention that post-hoc perform worse than all the rest including the baselines for both in- and out-domain.
Does that suggest post-hoc is not suitable for detection tasks for some reasons?

---

> ### Author Response · Authors · 2022-08-02
> **Response to Reviewer GHp5**
>
> Thank you for the positive comments and an interesting question. In Table-1, where we present results based on FCOS (a single-stage detector), post-hoc performs inferior to the baseline. This, however, is not true for Table-2 where Faster-RCNN (a two-stage detector) is being used. Post-hoc method relies on finding a single temperature T value to scale the logits. In detection architectures e.g., single-stage detectors (FCOS), where we have multiple dense location-dependent output maps, it is likely that the single T value will be sub-optimal for the logit vectors corresponding to a large number of output locations.  The architecture-sensitive performance variations of post-hoc temperature scaling are also observed in classification applications (e.g., see Table 1 in ref. [21] of the main paper).

---

> > ### Comment · Reviewer_GHp5 · 2022-08-07
> > **Final Comment**
> >
> > Thanks for addressing my concerns. The problem has been solved. I decide to keep the score.

---

### Meta-Review · Area_Chair_W1i4 · 2022-08-31

**Recommendation:** Accept
**Confidence:** Certain

**Metareview:**

This paper proposes a calibration loss function and uncertainty quantification mechanism to improve object detection calibration for both in-domain and out-of-domain detections. The reviewers have found this paper novel and clearly written with strong empirical results. Given this, I am happy to recommend this paper for acceptance at NeurIPS.

**Award:**

No

---

### Decision · Program_Chairs · 2022-09-14

Accept